# Scalable Inference for Logistic-Normal Topic Models

**Jianfei Chen, Jun Zhu, Zi Wang, Xun Zheng and Bo Zhang**
State Key Lab of Intelligent Tech. & Systems; Tsinghua National TNList Lab;
Department of Computer Science and Technology, Tsinghua University, Beijing 100084, China
{chenjf10,wangzi10}@mails.tsinghua.edu.cn;
{dcszj,dcszb}@mail.tsinghua.edu.cn; xunzheng@cs.cmu.edu

## Abstract

Logistic-normal topic models can effectively discover correlation structures among latent topics. However, their inference remains a challenge because of the non-conjugacy between the logistic-normal prior and multinomial topic mixing proportions. Existing algorithms either make restricting mean-field assumptions or are not scalable to large-scale applications. This paper presents a partially collapsed Gibbs sampling algorithm that approaches the provably correct distribution by exploring the ideas of data augmentation. To improve time efficiency, we further present a parallel implementation that can deal with large-scale applications and learn the correlation structures of thousands of topics from millions of documents. Extensive empirical results demonstrate the promise.

## 1    Introduction

In Bayesian models, though conjugate priors normally result in easier inference problems, non-conjugate priors could be more expressive in capturing desired model properties. One popular example is admixture topic models which have obtained much success in discovering latent semantic structures from data. For the most popular latent Dirichlet allocation (LDA) [5], a Dirichlet distribution is used as the conjugate prior for multinomial mixing proportions. But a Dirichlet prior is unable to model topic correlation, which is important for understanding/visualizing the semantic structures of complex data, especially in large-scale applications. One elegant extension of LDA is the logistic-normal topic models (aka *correlated topic models*, CTMs) [3], which use a logistic-normal prior to capture the correlation structures among topics effectively. Along this line, many subsequent extensions have been developed, including dynamic topic models [4] that deal with time series via a dynamic linear system on the Gaussian variables and infinite CTMs [11] that can resolve the number of topics from data.

The modeling flexibility comes with computational cost. Although significant progress has been made on developing scalable inference algorithms for LDA using either distributed [10, 16, 1] or online [7] learning methods, the inference of logistic-normal topic models still remains a challenge, due to the non-conjugate priors. Existing algorithms on learning logistic-normal topic models mainly rely on approximate techniques, e.g., variational inference with unwarranted mean-field assumptions [3]. Although variational methods have a deterministic objective to optimize and are usually efficient, they could only achieve an approximate solution. If the mean-field assumptions are not made appropriately, the approximation could be unsatisfactory. Furthermore, existing algorithms can only deal with small corpora and learn a limited number of topics. It is important to develop scalable algorithms in order to apply the models to large collections of documents, which are becoming increasingly common in both scientific and engineering fields.

To address the limitations listed above, we develop a scalable Gibbs sampling algorithm for logistic-normal topic models, without making any restricting assumptions on the posterior distribution. Technically, to deal with the non-conjugate logistic-normal prior, we introduce auxiliary Polya-Gamma

variables [13], following the statistical ideas of data augmentation [17, 18, 8]; and the augmented posterior distribution leads to conditional distributions from which we can draw samples easily without accept/reject steps. Moreover, the auxiliary variables are locally associated with each individual document, and this locality naturally allows us to develop a distributed sampler by splitting the documents into multiple subsets and allocating them to multiple machines. The global statistics can be updated asynchronously without sacrificing the predictive ability on unseen testing documents. We successfully apply the scalable inference algorithm to learning a correlation graph of thousands of topics on large corpora with millions of documents. These results are the largest *automatically learned* topic correlation structures to our knowledge.

## 2 Logistic-Normal Topic Models

Let $\mathbf{W} = \{\mathbf{w}_d\}_{d=1}^D$ be a set of documents, where $\mathbf{w}_d = \{w_{dn}\}_{n=1}^{N_d}$ denote the words appearing in document $d$ of length $N_d$. A hierarchical Bayesian topic model posits each document as an admixture of $K$ topics, where each topic $\mathbf{\Phi}_k$ is a multinomial distribution over a $V$-word vocabulary. For a logistic-normal topic model (e.g., CTM), the generating process of document $d$ is:

$$\boldsymbol{\eta}_d \sim \mathcal{N}(\boldsymbol{\mu}, \mathbf{\Sigma}), \quad \theta_d^k = \frac{e^{\eta_d^k}}{\sum_{j=1}^K e^{\eta_d^j}}, \forall n \in \{1, \cdots, N_d\} : z_{dn} \sim \mathrm{Mult}(\boldsymbol{\theta}_d), \ w_{dn} \sim \mathrm{Mult}(\mathbf{\Phi}_{z_{dn}}),$$

where $\mathrm{Mult}(\cdot)$ denotes the multinomial distribution; $z_{dn}$ is a $K$-binary vector with only one nonzero element; and $\mathbf{\Phi}_{z_{dn}}$ denotes the topic selected by the non-zero entry of $z_{dn}$. For Bayesian CTM, the topics are samples drawn from a prior, e.g., $\mathbf{\Phi}_k \sim \mathrm{Dir}(\boldsymbol{\beta})$, where $\mathrm{Dir}(\cdot)$ is a Dirichlet distribution. Note that for identifiability, normally we assume $\eta_d^K = 0$.

Given a set of documents $\mathbf{W}$, CTM infers the posterior distribution $p(\boldsymbol{\eta}, \mathbf{Z}, \mathbf{\Phi}|\mathbf{W}) \propto p_0(\boldsymbol{\eta}, \mathbf{Z}, \mathbf{\Phi})p(\mathbf{W}|\mathbf{Z}, \mathbf{\Phi})$ by the Bayes' rule. This problem is generally hard because of the non-conjugacy between the normal prior and the logistic transformation function (can be seen as a likelihood model for $\boldsymbol{\theta}$). Existing approaches resort to variational approximate methods [3] with strict factorization assumptions. To avoid mean-field assumptions and improve the inference accuracy, below we present a partially collapsed Gibbs sampler, which is simple to implement and can be naturally parallelized for large-scale applications.

## 3 Gibbs Sampling with Data Augmentation

We now present a block-wise Gibbs sampling algorithm for logistic-normal topic models. To improve mixing rates, we first integrate out the Dirichlet variables $\mathbf{\Phi}$, by exploring the conjugacy between a Dirichlet prior and multinomial likelihood. Specifically, we can integrate out $\mathbf{\Phi}$ and perform Gibbs sampling for the marginalized distribution:

$$p(\boldsymbol{\eta}, \mathbf{Z}|\mathbf{W}) \propto p(\mathbf{W}|\mathbf{Z})\prod_{d=1}^D\Big(\prod_{n=1}^{N_d}\theta_d^{z_{dn}}\Big)\mathcal{N}(\boldsymbol{\eta}_d|\boldsymbol{\mu}, \mathbf{\Sigma}) \propto \prod_{k=1}^K\frac{\delta(\mathbf{C}_k + \boldsymbol{\beta})}{\delta(\boldsymbol{\beta})}\prod_{d=1}^D\Big(\prod_{n=1}^{N_d}\frac{e^{\eta_d^{z_{dn}}}}{\sum_{j=1}^K e^{\eta_d^j}}\Big)\mathcal{N}(\boldsymbol{\eta}_d|\boldsymbol{\mu}, \mathbf{\Sigma}),$$

where $C_k^t$ is the number of times topic $k$ being assigned to the term $t$ over the whole corpus; $\mathbf{C}_k = \{C_k^t\}_{t=1}^V$; and $\delta(\mathbf{x}) = \frac{\prod_{i=1}^{\dim(\mathbf{x})}\Gamma(x_i)}{\Gamma(\sum_{i=1}^{\dim(\mathbf{x})}x_i)}$ is a function defined with the Gamma function $\Gamma(\cdot)$.

### 3.1 Sampling Topic Assignments

When the variables $\boldsymbol{\eta} = \{\boldsymbol{\eta}_d\}_{d=1}^D$ are given, we draw samples from $p(\mathbf{Z}|\boldsymbol{\eta}, \mathbf{W})$. In our Gibbs sampler, this is done by iteratively drawing a sample for each word in each document. The local conditional distribution is:

$$p(z_{dn}^k = 1|\mathbf{Z}_{\neg n}, w_{dn}, \mathbf{W}_{\neg dn}, \boldsymbol{\eta}) \propto p(w_{dn}|z_{dn}^k = 1, \mathbf{Z}_{\neg n}, \mathbf{W}_{\neg dn})e^{\eta_d^k} \propto \frac{C_{k, \neg n}^{w_{dn}} + \beta_{w_{dn}}}{\sum_{j=1}^V C_{k, \neg n}^j + \sum_{j=1}^V \beta_j}e^{\eta_d^k}, (1)$$

where $C_{\cdot, \neg n}^{\cdot}$ indicates that term $n$ is excluded from the corresponding document or topic.

### 3.2 Sampling Logistic-Normal Parameters

When the topic assignments $\mathbf{Z}$ are given, we draw samples from the posterior distribution $p(\boldsymbol{\eta}|\mathbf{Z}, \mathbf{W}) \propto \prod_{d=1}^D\left(\prod_{n=1}^{N_d}\frac{e^{\eta_{z_n}^d}}{\sum_{j=1}^K e^{\eta_j^d}}\right)\mathcal{N}(\boldsymbol{\eta}_d|\boldsymbol{\mu}, \mathbf{\Sigma})$, which is a Bayesian logistic regression model

with $\mathbf{Z}$ as the multinomial observations. Though it is hard to draw samples directly due to non-conjugacy, we can leverage recent advances in data augmentation to solve this inference task efficiently, with analytical local conditionals for Gibbs sampling, as detailed below.

Specifically, we have the likelihood of "observing" the topic assignments $\mathbf{z}_d$ for document $d$ [1] as $p(\mathbf{z}_d|\boldsymbol{\eta}_d) = \prod_{n=1}^{N_d} \frac{e^{\eta_d^{z_{dn}}}}{\sum_{j=1}^{K} e^{\eta_d^j}}$. Following Homes & Held [8], the likelihood for $\eta_k^d$ conditioned on $\boldsymbol{\eta}_d^{\neg k}$ is:

$$\ell(\eta_d^k|\boldsymbol{\eta}_d^{\neg k}) = \prod_{n=1}^{N_d} \Big( \frac{e^{\rho_d^k}}{1+e^{\rho_d^k}} \Big)^{z_{dn}^k} \Big( \frac{1}{1+e^{\rho_d^k}} \Big)^{1-z_{dn}^k} = \frac{(e^{\rho_d^k})^{C_d^k}}{(1+e^{\rho_d^k})^{N_d}},$$

where $\rho_d^k = \eta_d^k - \zeta_d^k$; $\zeta_d^k = \log(\sum_{j\neq k} e^{\eta_d^j})$; and $C_d^k = \sum_{n=1}^{N_d} z_{dn}^k$ is the number of words assigned to topic $k$ in document $d$. Therefore, we have the conditional distribution

$$p(\eta_d^k|\boldsymbol{\eta}_d^{\neg k}, \mathbf{Z}, \mathbf{W}) \propto \ell(\eta_d^k|\boldsymbol{\eta}_d^{\neg k})\mathcal{N}(\eta_d^k|\mu_d^k, \sigma_k^2), \qquad (2)$$

where $\mu_d^k = \mu_k - \boldsymbol{\Lambda}_{kk}^{-1}\boldsymbol{\Lambda}_{k\neg k}(\boldsymbol{\eta}_d^{\neg k} - \boldsymbol{\mu}_{\neg k})$ and $\sigma_k^2 = \boldsymbol{\Lambda}_{kk}^{-1}$. $\boldsymbol{\Lambda} = \boldsymbol{\Sigma}^{-1}$ is the precision matrix of a Gaussian distribution.

This is a posterior distribution of a Bayesian logistic model with a Gaussian prior, where $z_{dn}^k$ are binary response variables. Due to the non-conjugacy between the normal prior and logistic likelihood, we do not have analytical form of this posterior distribution. Although standard Monte Carlo methods (e.g., rejection sampling) can be applied, they normally require a good proposal distribution and may have the trouble to deal with accept/reject rates. Data augmentation techniques have been developed, e.g., [8] presented a two layer data augmentation representation with logistic distributions and [9] applied another data augmentation with uniform variables and truncated Gaussian distributions, which may involve sophisticated accept/reject strategies [14]. Below, we develop a simple exact sampling method without a proposal distribution.

Our method is based on a new data augmentation representation, following the recent developments in Bayesian logistic regression [13], which is a direct data augmentation scheme with only one layer of auxiliary variables and does not need to tune in order to get optimal performance. Specifically, for the above posterior inference problem, we can show the following lemma.

**Lemma 1** (Scale Mixture Representation). *The likelihood $\ell(\eta_d^k|\boldsymbol{\eta}_d^{\neg k})$ can be expressed as*

$$\frac{(e^{\rho_d^k})^{C_d^k}}{(1+e^{\rho_d^k})^{N_d}} = \frac{1}{2^{N_d}} e^{\kappa_d^k \rho_d^k} \int_0^\infty e^{-\frac{\lambda_d^k(\rho_d^k)^2}{2}} p(\lambda_d^k|N_d, 0)d\lambda_d^k,$$

*where $\kappa_d^k = C_d^k - N_d/2$ and $p(\lambda_d^k|N_d, 0)$ is the Polya-Gamma distribution $\mathcal{PG}(N_d, 0)$.*

The lemma suggest that $p(\eta_d^k|\boldsymbol{\eta}_d^{\neg k}, \mathbf{Z}, \mathbf{W})$ is a marginal distribution of the complete distribution

$$p(\eta_d^k, \lambda_d^k|\boldsymbol{\eta}_d^{\neg k}, \mathbf{Z}, \mathbf{W}) \propto \frac{1}{2^{N_d}} \exp\Big( \kappa_d^k \rho_d^k - \frac{\lambda_d^k(\rho_d^k)^2}{2}\Big) p(\lambda_d^k|N_d, 0)\mathcal{N}(\eta_d^k|\mu_d^k, \sigma_k^2).$$

Therefore, we can draw samples from the complete distribution. By discarding the augmented variable $\lambda_d^k$, we get the samples of the posterior distribution $p(\eta_d^k|\boldsymbol{\eta}_d^{\neg k}, \mathbf{Z}, \mathbf{W})$.

**For $\eta_d^k$:** we have $p(\eta_d^k|\boldsymbol{\eta}_d^{\neg k}, \mathbf{Z}, \mathbf{W}, \lambda_d^k) \propto \exp\big( \kappa_d^k \eta_d^k - \frac{\lambda_d^k(\eta_d^k)^2}{2}\big)\mathcal{N}(\eta_d^k|\mu, \sigma^2) = \mathcal{N}(\gamma_d^k, (\tau_d^k)^2)$, where the posterior mean is $\gamma_d^k = (\tau_d^k)^2(\sigma_k^{-2}\mu_d^k + \kappa_d^k + \lambda_d^k\zeta_d^k)$ and the variance is $(\tau_d^k)^2 = (\sigma_k^{-2} + \lambda_d^k)^{-1}$. Therefore, we can easily draw a sample from a univariate Gaussian distribution.

**For $\lambda_d^k$:** the conditional distribution of the augmented variable is $p(\lambda_d^k|\mathbf{Z}, \mathbf{W}, \boldsymbol{\eta}) \propto \exp\big( -\frac{\lambda_d^k(\rho_d^k)^2}{2}\big)p(\lambda_d^k|N_d, 0) = \mathcal{PG}\big(\lambda_d^k; N_d, \rho_d^k\big)$, which is again a Polya-Gamma distribution by using the construction definition of the general $\mathcal{PG}(a, b)$ class through an exponential tilting of the $\mathcal{PG}(a, 0)$ density [13]. To draw samples from the Polya-Gamma distribution, note that a naive implementation of the sampling using the infinite sum-of-Gamma representation is not efficient and it also involves a potentially inaccurate step of truncating the infinite sum. Here we adopt the exact method proposed in [13], which draws the samples through drawing $N_d$ samples from $\mathcal{PG}(1, \eta_d^k)$. Since $N_d$ is normally large, we will develop a fast and effective approximation in the next section.

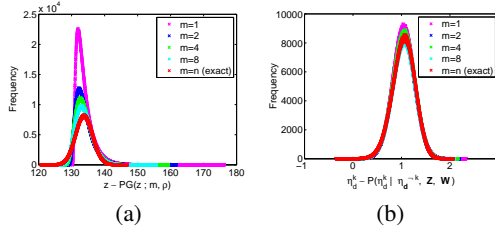

Figure 3: (a) frequency of $f(z)$ with $z \sim \mathcal{PG}(m, \rho)$; and (b) frequency of samples from $\eta_d^k \sim p(\eta_d^k | \boldsymbol{\eta}_d^{-k}, \mathbf{Z}, \mathbf{W})$. Though $z$ is not from the exact distribution, the distribution of $\eta_d^k$ is very accurate. The parameters $\rho_d^k = -4.19, C_d^k = 19, N_d = 1155, \mu_d^k = 0.40, \sigma_d^2 = 0.31$, and $\zeta = 5.35$ are from a real distribution when training on the NIPS data set.

### 3.3 Fully-Bayesian Models

We can treat $\boldsymbol{\mu}$ and $\boldsymbol{\Sigma}$ as random variables and perform fully-Bayesian inference, by using the conjugate Normal-Inverse-Wishart prior, $p_0(\boldsymbol{\mu}, \boldsymbol{\Sigma}) = \mathcal{NIW}(\boldsymbol{\mu}_0, \rho, \kappa, W)$, that is

$$\boldsymbol{\Sigma}|\kappa, W \sim \mathcal{IW}(\boldsymbol{\Sigma}; \kappa, W^{-1}), \ \boldsymbol{\mu}|\boldsymbol{\Sigma}, \boldsymbol{\mu}_0, \rho \sim \mathcal{N}(\boldsymbol{\mu}; \boldsymbol{\mu}_0, \boldsymbol{\Sigma}/\rho),$$

where $\mathcal{IW}(\boldsymbol{\Sigma}; \kappa, W^{-1}) = \frac{|W|^{\kappa/2}}{2^{\frac{\kappa M}{2}} \Gamma_M(\frac{\kappa}{2}) |\boldsymbol{\Sigma}|^{\frac{\kappa+M+1}{2}}} \exp(-\frac{1}{2}\text{Tr}(W\boldsymbol{\Sigma}^{-1}))$ is the inverse Wishart distribution and $(\boldsymbol{\mu}_0, \rho, \kappa, W)$ are hyper-parameters. Then, the conditional distribution is

$$p(\boldsymbol{\mu}, \boldsymbol{\Sigma}|\boldsymbol{\eta}, \mathbf{Z}, \mathbf{W}) \propto p_0(\boldsymbol{\mu}, \boldsymbol{\Sigma}) \prod_d p(\boldsymbol{\eta}_d|\boldsymbol{\mu}, \boldsymbol{\Sigma}) = \mathcal{NIW}(\boldsymbol{\mu}_0', \rho', \kappa', W'), \tag{3}$$

which is still a Normal-Inverse-Wishart distribution due to the conjugate property and the parameters are $\boldsymbol{\mu}_0' = \frac{\rho}{\rho+D}\boldsymbol{\mu}_0 + \frac{D}{\rho+D}\bar{\boldsymbol{\eta}}$, $\rho' = \rho + D$, $\kappa' = \kappa + D$ and $W' = W + Q + \frac{\rho D}{\rho+D}(\bar{\boldsymbol{\eta}} - \boldsymbol{\mu}_0)(\bar{\boldsymbol{\eta}} - \boldsymbol{\mu}_0)^\top$, where $\bar{\boldsymbol{\eta}} = \frac{1}{D}\sum_d \boldsymbol{\eta}_d$ is the empirical mean of the data and $Q = \sum_d (\boldsymbol{\eta}_d - \bar{\boldsymbol{\eta}})(\boldsymbol{\eta}_d - \bar{\boldsymbol{\eta}})^\top$.

## 4 Parallel Implementation and Fast Approximate Sampling

The above Gibbs sampler can be naturally parallelized to extract large correlation graphs from millions of documents, due to the following observations:

First, both $\boldsymbol{\eta}_d$ and $\boldsymbol{\lambda}_d$ are conditionally independent given $\boldsymbol{\mu}$ and $\boldsymbol{\Sigma}$, which makes it natural to distribute documents over machines and infer local $\boldsymbol{\eta}_d$ and $\boldsymbol{\lambda}_d$. No communication is needed for this sampling step. Second, the global variables $\boldsymbol{\mu}$ and $\boldsymbol{\Sigma}$ can be inferred and broadcast to every machine after each iteration. As mentioned in Section 3.3, this involves: 1) computing $\mathcal{NIW}$ posterior parameters, and 2) sampling from Eq. 3. Notice that $\boldsymbol{\eta}_d$ contribute to the posterior parameters $\boldsymbol{\mu}_0', W'$ through the simple summation operator, so that we can perform local summation on each machine, followed by a global aggregation. Similarly, $\mathcal{NIW}$ sample can be drawn distributively, by computing sample covariance of $x_1, \cdots, x_{\kappa'}$, drawn from $\mathcal{N}(x|0, W')$ distributively after broadcasting $W'$. Finally, the topic assignments $\mathbf{z}_d$ are conditionally independent given the topic counts $\mathbf{C}_k$. We synchronize $\mathbf{C}_k$ globally by leveraging the recent advances on scalable inference of LDA [1, 16], which implemented a general framework to synchronize such counts.

To further speed up the inference algorithm, we designed a fast approximate sampling method to draw $\mathcal{PG}(n, \rho)$ samples, reducing the time complexity from $O(n)$ in [13] to $O(1)$. Specifically, Polson et al. [13] show how to efficiently generate $\mathcal{PG}(1, \rho)$ random variates. Due to additive property of Polya-Gamma distribution, $y \sim \mathcal{PG}(n, \rho)$ if $x_i \sim \mathcal{PG}(1, \rho)$ and $y = \sum_{i=1}^n x_i$. However, this sampler can be slow when $n$ is large. For our Gibbs sampler, $n$ is the document length, often around hundreds. Fortunately, an effective approximation can be developed to achieve constant time sampling of $\mathcal{PG}$. Since $n$ is relatively large, the sum variable $y$ should be almost normally distributed, according to the central limit theorem. Fig. 3(a) confirms this intuition. Consider another PG variable $z \sim \mathcal{PG}(m, \rho)$. If both $m$ and $n$ are large, $y$ and $z$ should be both samples from normal distribution. Hence, we can do a simple linear transformation of $z$ to approximate $y$. Specifically, we have $f(z) = \sqrt{Var(y)/Var(z)}(z - \mathbb{E}[z]) + \mathbb{E}[y]$, where $\mathbb{E}[y] = \frac{n}{2\rho}tanh(\rho/2)$ from [12], and $\frac{Var(z)}{Var(y)} = \frac{m}{n}$ since both $y$ and $z$ are sum of $\mathcal{PG}(1, \rho)$ variates. It can be shown that $f(z)$ and $y$ have the same mean and variance. In practice, we found that even when $m = 1$, the algorithm still can draw good samples from $p(\eta_d^k | \boldsymbol{\eta}_d^{-k}, \mathbf{Z}, \mathbf{W})$ (See Fig. 3(b)). Hence, we are able to speed up the Polya-Gamma sampling process significantly by applying this approximation. More empirical analysis can be found in the appendix.

Furthermore, we can perform sparsity-aware fast sampling [19] in the Gibbs sampler. Specifically, let $A_k = \frac{C_{k,\neg n}^{w_{dn}}}{\sum_{j=1}^{V} C_{k,\neg n}^{j} + \sum_{j=1}^{V} \beta_j} e^{\eta_d^k}, B_k = \frac{\beta_{w_{dn}}}{\sum_{j=1}^{V} C_{k,\neg n}^{j} + \sum_{j=1}^{V} \beta_j} e^{\eta_d^k}$, then Eq. (1) can be written as $p(z_{dn}^k = 1|\mathbf{Z}_{\neg n}, w_{dn}, \mathbf{W}_{\neg dn}, \boldsymbol{\eta}) \propto A_k + B_k$. Let $Z_A = \sum_k A_k$ and $Z_B = \sum_k B_k$. We can show that the sampling of $z_{dn}$ can be done by sampling from $\mathrm{Mult}(\frac{A}{Z_A})$ or $\mathrm{Mult}(\frac{B}{Z_B})$, due to the fact:

$$p(z_{dn}^k = 1|\mathbf{Z}_{\neg n}, w_{dn}, \mathbf{W}_{\neg dn}, \boldsymbol{\eta}) = \frac{A_k}{Z_A + Z_B} + \frac{B_k}{Z_A + Z_B} = (1-p)\frac{A_k}{Z_A} + p\frac{B_k}{Z_B}, \qquad (4)$$

where $p = \frac{Z_B}{Z_A + Z_B}$. Note that Eq. (4) is a marginalization with respect to an auxiliary binary variable. Thus a sample of $z_{dn}$ can be drawn by flipping a coin with probability $p$ being head. If it is tail, we draw $z_{dn}$ from $\mathrm{Mult}(\frac{A}{Z_A})$; otherwise from $\mathrm{Mult}(\frac{B}{Z_B})$. The advantage is that we only need to consider all non-zero entries of $A$ to sample from $\mathrm{Mult}(\frac{A}{Z_A})$. In fact, $A$ has few non-zero entries due to the sparsity of the topic counts $\mathbf{C}_k$. Thus, the time complexity would be reduced from $O(K)$ to $O(s(K))$, where $s(K)$ is the average number of non-zero entries in $\mathbf{C}_k$. In practice, $\mathbf{C}_k$ is very sparse, hence $s(K) \ll K$ when $K$ is large. To sample from $\mathrm{Mult}(\frac{B}{Z_B})$, we iterate over all $K$ potential assignments. But since $p$ is typically small, $O(K)$ time complexity is acceptable.

With the above techniques, the time complexity per document of the Gibbs sampler is $O(N_d s(K))$ for sampling $\mathbf{z}_d$, $O(K^2)$ for computing $(\mu_d^k, \sigma_k^2)$, and $O(SK)$ for sampling $\boldsymbol{\eta}_d$ with Eq. (2), where $S$ is the number of sub-burn-in steps over sampling $\eta_d^k$. Thus the overall time complexity is $O(N_d s(K) + K^2 + SK)$, which is higher than the $O(N_d s(K))$ complexity of LDA [1] when $K$ is large, indicating a cost for the enriched representation of CTM comparing to LDA.

## 5 Experiments

We now present qualitative and quantitative evaluation to demonstrate the efficacy and scalability of the Gibbs sampler for CTM (denoted by gCTM). Experiments are conducted on a 40-node cluster, where each node is equipped with two 6-core CPUs (2.93GHz). For all the experiments, if not explicitly mentioned, we set the hyper-parameters as $\beta = 0.01$, $T = 350$, $S = 8$, $m = 1$, $\rho = \kappa = 0.01D$, $\boldsymbol{\mu}_0 = 0$, and $W = \kappa I$, where $T$ is the number of burn-in steps. We will use $M$ to denote the number of machines and $P$ to denote the number of CPU cores. For baselines, we compare with the variational CTM (vCTM) [3] and the state-of-the-art LDA implementation, Yahoo! LDA (Y!LDA) [1]. In order to achieve fair comparison, for both vCTM and gCTM we select $T$ such that the models converge sufficiently, as we shall discuss later in Section 5.3.

**Data Sets**: Experiments are conducted on several benchmark data sets, including NIPS paper abstracts, 20Newsgroups, and NYTimes (New York Times) corpora from [2] and the Wikipedia corpus from [20]. All the data sets are randomly split into training and testing sets. Following the settings in [3], we partition each document in the testing set into an observed part and a held-out part.

### 5.1 Qualitative Evaluation

We first examine the correlation structure of 1,000 topics learned by CTM using our scalable sampler on the NYTimes corpus with 285,000 documents. Since the entire correlation graph is too large, we build a 3-layer hierarchy by clustering the learned topics, with their learned correlation strength as the similarity measure. Fig. 4 shows a part of the hierarchy[2], where the subgraph $A$ represents the top layer with 10 clusters. The subgraphs $B$ and $C$ are two second layer clusters; and $D$ and $E$ are two correlation subgraphs consisting of leaf nodes (i.e., learned topics). To represent their semantic meanings, we present 4 most frequent words for each topic; and for each topic cluster, we also show most frequent words by building a *hyper-topic* that aggregates all the included topics. On the top layer, the font size of each word in a word cloud is proportional to its frequency in the hyper-topic. Clearly, we can see that many topics have strong correlations and the structure is useful to help humans understand/browse the large collection of topics. With 40 machines, our parallel Gibbs sampler finishes the training in 2 hours, which means that we are able to process real world corpus in considerable speed. More details on scalability will be provided below.

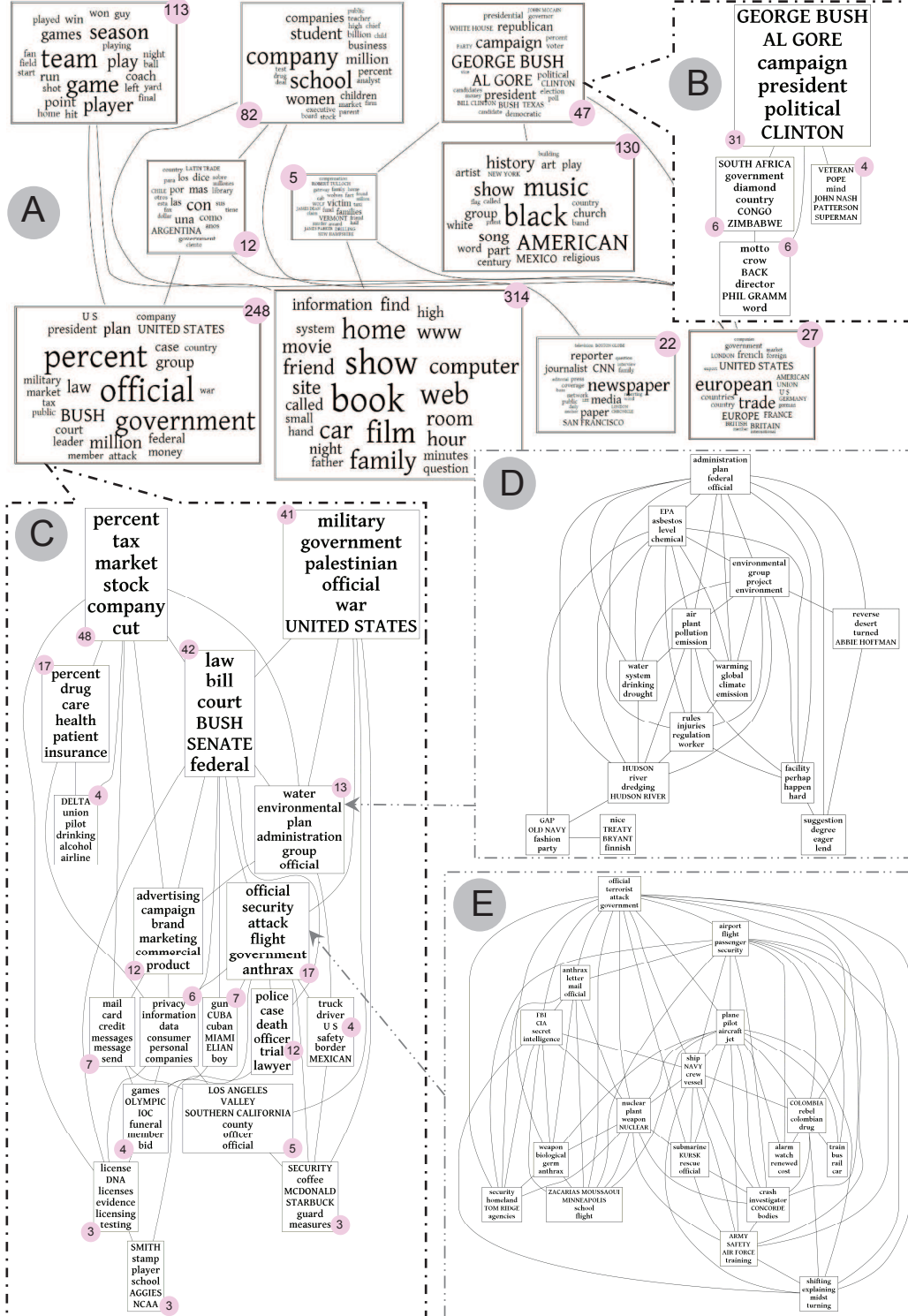

Figure 4: A hierarchical visualization of the correlation graph with 1,000 topics learned from 285,000 articles of the NYTimes. *A* denotes the top-layer subgraph with 10 big clusters; *B* and *C* denote two second-layer clusters; and *D* and *E* are two subgraphs with leaf nodes (i.e., topics). We present most frequent words of each topic cluster. Edges denote a correlation (above some threshold) and the distance between two nodes represents the strength of their correlation. The node size of a cluster is determined by the number of topics included in that cluster.

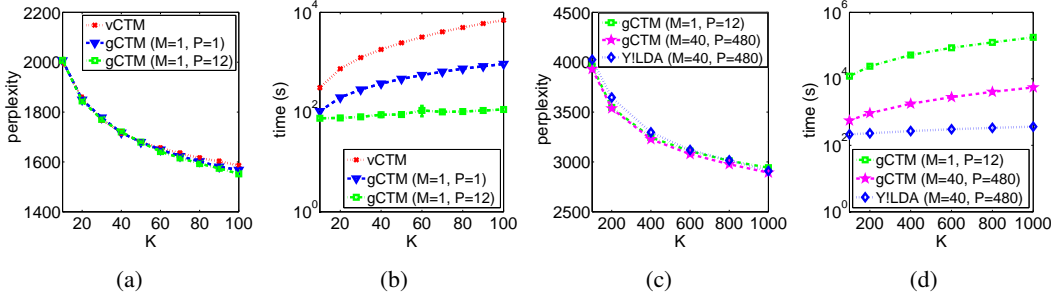

Figure 5: (a)(b): Perplexity and training time of vCTM, single-core gCTM, and multi-core gCTM on the NIPS data set; (c)(d): Perplexity and training time of single-machine gCTM, multi-machine gCTM, and multi-machine Y!LDA on the NYTimes data set.

## 5.2 Performance

We begin with an empirical assessment on the small NIPS data set, whose training set contains 1.2K documents. Fig. 5(a)&(b) show the performance of three single-machine methods: vCTM ($M = 1, P = 1$), sequential gCTM ($M = 1, P = 1$), and parallel gCTM ($M = 1, P = 12$). Fig. 5(a) shows that both versions of gCTM produce similar or better perplexity, compared to vCTM. Moreover, Fig. 5(b) shows that when $K$ is large, the advantage of gCTM becomes salient, e.g., sequential gCTM is about 7.5 times faster than vCTM; and multi-core gCTM achieves almost two orders of magnitude of speed-up compared to vCTM.

In Table 1, we compare the efficiency of vCTM and gCTM on different sized data sets. It can be observed that vCTM immediately becomes impractical when the data size reaches 285K, while by utilizing additional computing resources, gCTM is able to process larger data sets with considerable speed, making it applicable to real world problems. Note

| data set | $D$ | $K$ | vCTM | gCTM |
|----------|-----|-----|------|------|
| NIPS | 1.2K | 100 | 1.9 hr | 8.9 min |
| 20NG | 11K | 200 | 16 hr | 9 min |
| NYTimes | 285K | 400 | N/A* | 0.5 hr |
| Wiki | 6M | 1000 | N/A* | 17 hr |

*not finished within 1 week.

Table 1: Training time of vCTM and gCTM ($M = 40$) on various datasets.

that gCTM has almost the same training time on NIPS and 20Newsgroups data sets, due to their small sizes. In such cases, the algorithm is dominated by synchronization rather than computation.

Fig. 5(c)&(d) show the results on the NYTimes corpus, which contains over 285K training documents and cannot be handled well by non-parallel methods. Therefore we concentrate on three parallel methods — single-machine gCTM ($M = 1, P = 12$), multi-machine gCTM ($M = 40, P = 480$), and multi-machine Y!LDA ($M = 40, P = 480$). We can see that: 1) both versions of gCTM obtain comparable perplexity to Y!LDA; and 2) gCTM ($M = 40$) is over an order of magnitude faster than the single-machine method, achieving considerable speed-up with additional computing resources. These observations suggest that gCTM is able to handle large data sets without sacrificing the quality of inference. Also note that Y!LDA is faster than gCTM because of the model difference — LDA does not learn correlation structure among topics. As analyzed in Section 4, the time complexity of gCTM is $O(K^2 + SK + N_d s(K))$ per document, while for LDA it is $O(N_d s(K))$.

## 5.3 Sensitivity

**Burn-In and Sub-Burn-In**: Fig. 6(a)&(b) show the effect of burn-in steps and sub-burn-in steps on the NIPS data set with $K = 100$. We also include vCTM for comparison. For vCTM, $T$ denotes the number of iteration of its EM loop in variational context. Our main observations are twofold: 1) despite various $S$, all versions of gCTMs reach a similar level of perplexity that is better than vCTM; and 2) a moderate number of sub-iterations, e.g. $S = 8$, leads to the fastest convergence.

This experiment also provides insights on determining the number of outer iterations $T$ that assures convergence for both models. We adopt Cauchy's criterion [15] for convergence: given an $\epsilon > 0$, an algorithm converges at iteration $T$ if $\forall i, j \geq T$, $|Perp_i - Perp_j| < \epsilon$, where $Perp_i$ and $Perp_j$ are perplexity at iteration $i$ and $j$ respectively. In practice, we set $\epsilon = 20$ and run experiments with very large number of iterations. As a result, we obtained $T = 350$ for gCTM and $T = 8$ for vCTM, as pointed out with corresponding verticle line segments in Fig. 6(a)&(b).

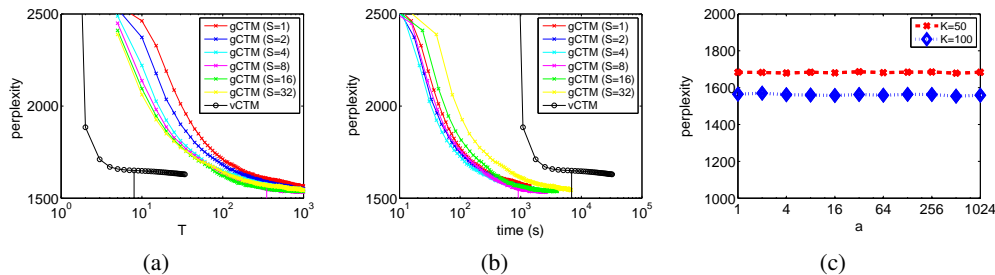

Figure 6: Sensitivity analysis with respect to key hyper-parameters: (a) perplexity at each iteration with different $S$; (b) convergence speed with different $S$; (c) perplexity tested with different prior.

**Prior**: Fig. 6(c) shows perplexity under different prior settings. To avoid expensive search in a huge space, we set $(\boldsymbol{\mu}_0, \rho, W, \kappa) = (0, a, aI, a)$ to test the effect of $\mathcal{NIW}$ prior, where a larger $a$ implies more pseudo-observations of $\boldsymbol{\mu} = 0, \boldsymbol{\Sigma} = I$. We can see that for both $K = 50$ and $K = 100$, the perplexity is invariant under a wide range of prior settings. This suggests that gCTM is insensitive to prior values.

## 5.4  Scalability

Fig. 7 shows the scalability of gCTM on the large Wikipedia data set with $K = 500$. A practical problem in real world machine learning is that when computing resources are limitted, as the data size grows, the running time soon upsurges to an untolerable level. Ideally, this problem can be solved by adding the same ratio of computing nodes. Our experiment demonstrates that gCTM performs well in this scenario — as we pour in the same proportion of data and machines, the training time is almost kept constant. In fact, the largest difference from ideal curve is about 1,000 seconds, which is almost unobservable in the figure. This suggests that parallel gCTM enjoys nice scalability.

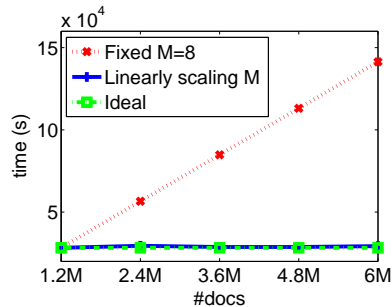

Figure 7: Scalability analysis. We set $M = 8, 16, 24, 32, 40$ so that each machine processes 150K documents.

## 6  Conclusions and Discussions

We present a scalable Gibbs sampling algorithm for logistic-normal topic models. Our method builds on a novel data augmentation formulation and addresses the non-conjugacy without making strict mean-field assumptions. The algorithm is naturally parallelizable and can be further boosted by approximate sampling techniques. Empirical results demonstrate significant improvement in time efficiency over existing variational methods, with slightly better perplexity. Our method enjoys good scalability, suggesting the ability to extract large structures from massive data.

In the future, we plan to study the performance of Gibbs CTM on industry level clusters with thousands of machines. We are also interested in developing scalable sampling algorithms of other logistic-normal topic models, e.g., infinite CTM and dynamic topic models. Finally, the fast sampler of Poly-Gamma distributions can be used in relational and supervised topic models [6, 21].

## Acknowledgments

This work is supported by the National Basic Research Program (973 Program) of China (Nos. 2013CB329403, 2012CB316301), National Natural Science Foundation of China (Nos. 61322308, 61305066), Tsinghua University Initiative Scientific Research Program (No. 20121088071), and Tsinghua National Laboratory for Information Science and Technology, China.

## Footnotes

[1] Due to the independence, we can treat documents separately.

[2]The entire correlation graph can be found on `http://ml-thu.net/~scalable-ctm`

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
