[Supplementary Material]

# Appendix

# 1 Sampling from Polya-Gamma Distribution

A random variable $X$ has a Polya-Gamma distribution with parameters $a > 0$ and $c \in \mathbb{R}$, if

$$X \overset{\mathrm{D}}{=} \frac{1}{2\pi^2} \sum_{k=1}^{\infty} \frac{g_k}{(k-1/2)^2 + c^2/(4\pi^2)} \tag{1}$$

where $g_k \sim Ga(a,1)$ are gamma random variables. By computing the truncated sum of Eq. 1, we can obtain a approximate sampler

$$X_{truncated} = \frac{1}{2\pi^2} \sum_{k=1}^{K} \frac{g_k}{(k-1/2)^2 + c^2/(4\pi^2)} \tag{2}$$

however, this approximation sampler is biased. [1] proposed a sampler which corrects the bias by multipling a constant

$$X_{\texttt{truncated}} = \frac{\mathbb{E}[X]}{\mathbb{E}[X_{truncated}]} \tag{3}$$

where $\mathbb{E}[X] = \frac{a}{2c} \tanh(\frac{c}{2})$ and $\mathbb{E}[X_{truncated}] = \frac{1}{2\pi^2} \sum_{k=1}^{K} \frac{a}{(k-1/2)^2 + c^2/(4\pi^2)}$, according to [3, 1]. Denote this approach as $\texttt{truncated}_K$.

[4] proposed a precise sampling algorithm for Polya-Gamma distributions

$$X_{\texttt{precise}} \overset{\mathrm{D}}{=} \sum_{n=1}^{a} X_n \tag{4}$$

where $X_n \sim PG(1,c)$ are i.i.d. samples. Denote this approach of $\texttt{precise}$. Draw samples from $PG(1,c)$ can be done in $O(1)$.[4]. However, $a$ is document length $N_d$ in logistic-normal topic models, since $N_d$ is quite large, $O(N_d)$ sampler is too slow. In this paper we draw $K << a$ samples instead. Denote this approach as $\texttt{pg1}_K$, note that $\texttt{pg1}_K = \texttt{precise}$.

Notes that $a = N_d$ is large, $X$ is sum of i.i.d. random variables. There is another approximation by the central limit theorem

$$X_{\texttt{gaussian}} \sim \mathcal{N}(\mu, \sigma^2) \tag{5}$$

Table 1: Comparison for different PG samplers.

| method | precise distribution? | precise mean? | precise variance? | time complexity |
|---|---|---|---|---|
| $\texttt{truncated}_K$ | no | yes | no | $O(K)$ |
| $\texttt{precise}$ | yes | yes | yes | $O(a)$ |
| $\texttt{pg1}_K$ | no | yes | yes | $O(K)$ |
| $\texttt{gaussian}$ | no | yes | yes | $O(1)$ |

where $\mu = \mathbb{E}[X], \sigma^2 = \mathrm{Var}[X]$. [3] has shown the moment-generating function of $PG(a,c)$

$$f(t) = \mathbb{E}[\exp(Xt)] = \frac{\cosh^a(c/2)}{\cosh^a(\frac{\sqrt{c^2-2t}}{2})} \tag{6}$$

we have

$$\mathbb{E}[X] = \lim_{t \to 0} f'(t) \tag{7}$$

$$= \frac{a}{2c} \tanh(\frac{c}{2}) \tag{8}$$

$$\mathbb{E}[X^2] = \lim_{t \to 0} f''(t) \tag{9}$$

$$= \frac{a(-(2+a)c^2 + ac^2\cosh(c) + 2c\sinh(c))}{8c^4\cosh(\frac{c}{2})^2} \tag{10}$$

and $\mathrm{Var}[X] = \mathbb{E}[X^2] - \mathbb{E}[X]^2$. Denote this as $\texttt{gaussian}$.

We summarize the algorithms mentioned above in Table 1. To compare these results, we draw 1,000,000 samples with different methods from $P(\lambda_d^k|\mathbf{Z}, \mathbf{W}, \boldsymbol{\eta})$, and use these samples to compute $P(\eta_d^k|\boldsymbol{\eta}_d\neg k, \mathbf{Z}, \mathbf{W})$. We compared their mean, variance and Kolmogorov-Smirnoff statistic, which is a measure of two empirical distributions $F_1(x)$ and $F_2(x)$: $KS(F_1(x), F_2(x)) = \max_x |F_1(x) - F_2(x)|$. Table 3 shows the result. We found in term of $KS(\eta)$, $\texttt{gaussian}$ did good, and $\texttt{truncated}_4$ performs similar with $\texttt{pg1}_1$. $\texttt{gaussian}$ is 4x faster than $\texttt{pg1}_1$, which is 2x faster than $\texttt{truncated}_4$.

Fig. 1 show the perplexity and time result on the real NIPS data set. We have similar observations: $\texttt{truncated}_K(K > 4)$ performs similar with $\texttt{pg1}_1$ and $\texttt{gaussian}$, but the latter two are faster. For larger data sets like NYTimes and 1,000 topics, we observed performance of $\texttt{pg1}_1$ and $\texttt{gaussian}$ are still similar, but $\texttt{truncated}_K$ suffer from numeral instablies: the sampled $\eta$ is getting to infinity and program crashes when $K < 32$. We think this instablies attributes to the imprecise variance. Both the performance and running time of $\texttt{truncated}_{32}$ are much worse than $\texttt{pg1}_1$ and $\texttt{gaussian}$. (Table 2)

## 2 More Sensitivity Results

We redo sensitivity analysis on a NYTimes data set while keep other experiment settings same as that in Section 5.3. We observed a plateau of the perplexity

Table 2: Comparison for different PG samplers on NYTimes corpus ($K = 1,000$).

| method | perplexity | time/s |
|---|---|---|
| $\mathtt{pg1}_1$ | 2913 | 5519 |
| $\mathtt{gaussian}$ | 2914 | 3984 |
| $\mathtt{truncated}_{32}$ | 2984 | 16270 |

Table 3: Comparison for different PG samplers. Parameters are same as Fig. 1 in the paper.

| method | $m$ | samples/second | $\mathrm{Var}[\lambda]$ | $\mathrm{KS}(\lambda)$ | $\mathbb{E}[\eta]$ | $\mathrm{KS}(\eta)$ |
|---|---|---|---|---|---|---|
| precise | - | 1,602 | 6.65 | - | 1.0459 | - |
| pg1 | 1 | 1,449,280 | 6.63 | 0.1146 | 1.0450 | 0.0146 |
| pg1 | 2 | 757,576 | 6.66 | 0.0810 | 1.0467 | 0.0088 |
| pg1 | 4 | 400,000 | 6.65 | 0.0562 | 1.0454 | 0.0080 |
| pg1 | 8 | 215,517 | 6.67 | 0.0391 | 1.0463 | 0.0051 |
| pg1 | 16 | 111,139 | 6.67 | 0.0259 | 1.0461 | 0.0041 |
| pg1 | 32 | 56,721 | 6.66 | 0.0176 | 1.0450 | 0.0055 |
| pg1 | 64 | 28,769 | 6.65 | 0.0123 | 1.0450 | 0.0049 |
| truncated | 1 | 3,846,150 | 15.49 | 0.1024 | 1.0241 | 0.0732 |
| truncated | 2 | 2,127,660 | 10.45 | 0.0558 | 1.0371 | 0.0350 |
| truncated | 4 | 1,111,110 | 8.37 | 0.0281 | 1.0415 | 0.0174 |
| truncated | 8 | 578,035 | 7.44 | 0.0140 | 1.0429 | 0.0087 |
| truncated | 16 | 313,480 | 7.04 | 0.0076 | 1.0441 | 0.0044 |
| truncated | 32 | 165,289 | 6.84 | 0.0039 | 1.0437 | 0.0043 |
| truncated | 64 | 84,962 | 6.76 | 0.0027 | 1.0449 | 0.0026 |
| gaussian | - | 6,250,000 | 6.66 | 0.0036 | 1.0458 | 0.0024 |

(a)  (b)

Figure 1: Perplexity and training time with different number of samples $m$.

Figure 2: Sensitivity analysis with respect to difference prior strength $a$.

Figure 3: Convergence speed for different number of subiterations $S$. (a)$K = 200$; (b)$K = 1000$.

when the number of pseudo-observations $a \in [10^3, 10^5]$ (Fig. 2), which corresponds to $[0.0035, 0.3509]$ of the number of training documents $D = 285,000$. This again showed the performance of our algorithm is not sensitivity to $a$. Sensitivity with respect to number of subiterations $S$ is howed in Fig. 3, we found the $S = 8$ sampler still converges fastest. This result is same as that on the small NIPS corpus. In conclusion, hyper parameters are relatively insensitive with respect to corpus size and number of topics, hyper parameters suggested in the paper ($a = 0.01D, S = 8$) are safe enough to use without tuning.

# 3 Comparison to Other Data Augmentation Algorithms

We compare our method with [2], who use a uniform distribution for data augmentation on the NIPS data set. By training $K = 100$ topics on the NIPS dataset, we found $S = 16$ leads to the fastest convergence for [2] (Fig. 4). Fig. 5 shows the perplexity and time consumption of our approach and [2], our ap-

Figure 4: Sensitivity analysis with respect to different number of subiterations. PG: our Polya-Gamma data augmentation approach. U: Uniform data augmentation approach [2].

proach is both more accurate and faster.

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

Figure 5: (a) Perplexity and (b) time for two algorithms on the NIPS corpus.