[Reviews · NeurIPS 2013]

Submitted by Assigned_Reviewer_4

The authors describe a two novel inference methods for the
correlated topic model (CTM). They build on analytic results for the
conditional logistic normal likelihood to arrive at a fast,
easily parallelized exact inference. This leads to an approximate
sampling method for producing Polya-Gamma variates. Finally, they
propose a method for efficiently drawing samples in the presence of
sparsity.

To demonstrate the utility of the topic correlations inferred
by the correlated topic model, the authors show a hierarchical
clustering of 1000 topics learned from the 300k document New
York Times dataset. While the figure makes a good case for the
usefulness of these sort of correlations, it seems a bit out of place
in the present paper, where the focus is on speeding up inference of an existing model as opposed to a suggestion of a new model. This page of the paper could have been better used further expanding on the details
of the inference.

To assess the effectiveness of their inference, the
authors compare against a variational CTM inference on the New
York Times dataset. The authors examine perplexity and training time
of the variational method and both serial and parallel runs of their
inference, in both cases finding little degradation in
perplexity performance while showing up to two orders of magnitude
faster training times.

To show the scalability of the approach, the authors compare the
training time and perplexity of their approach with that of
Y!LDA, both running on 480 distributed processors. Here they
demonstrate similar perplexity and while Y!LDA significantly
outperforms CTM in training runtime, this must be weighed against the
value of the correlation structure that CTM produces.

The authors give a brief discussion to the effect of the
various burn-in times of the inference and show that the priors have
no measurable effect on perplexity. That being said, this is
demonstrated only over one small corpus with 100 topics. How robust are these parameters to other corpus sizes and numbers of topics?

Finally, the authors note that the training time of their
parallel inference is nearly constant as document count and number of
compute nodes are increased in proportion. This is a clear
strength of the method.

The authors put forth a compelling inference for a challenging
model. They demonstrate prediction performance comparable to existing,
but much slower, variational methods while significantly improving
runtime. While the discussion of the inference itself could be
expanded (especially in light of the page spent on Figure 3),
the contribution is clear and enables the CTM to be used on
datasets far larger than previously feasible.

This paper comprises a solid contribution to the topic modeling community. So far, the slow inference of CTMs have prevented the community from building on it. This paper has the potential to change this. The derivation is sound and the evaluation is convincing.


--------------------
Update of comments after the discussion period:

In my understanding Y!LDA refers to a non-correlated version of LDA. I was impressed that this model is able to learn this more complicated model for a computational price that is comparable to Y!LDA, which is heavily tuned and less expressive.

I prefer papers that attempt a new approach, even if they do not strictly outperform a heavily tuned baseline. I think this approach is worthwhile considering and may lead to further improvements as people pick the idea up and refine it.


The fact that neither me nor Assigned_Reviewer_7 was not familiar the Polya-Gamma distribution is a sign that this knowledge should be promoted in the community, even if it is a straight forward application of work on Polya Gamma distributions. I would appreciate if relations to the paper of Polson, Scott and Windle, 2013 would be clarified in the writeup.


I have not seen many practical applications of the correlated topic model, and I think this is partially due to problems with the slow inference. I am hoping that this paper will motivate some new inference methods for this and other complex topic models, and that's why I would like to see it at the conference.


Summary: This paper comprises a solid contribution to the topic modeling community. So far, the slow inference of CTMs have prevented the community from building on it. This paper has the potential to change this. The derivation is sound and the evaluation is convincing.

Submitted by Assigned_Reviewer_6

The paper applies the recently proposed Polya-Gamma data augmentation techniques to derive a partially collapsed Gibbs sampling algorithm for the logistic-normal topic model. Although the paper appears to be the combination of two existing techniques and is not particularly novel, the proposed algorithm makes the inference for logistic-normal topic models scalable to large scale applications.

Quality and Clarity:

The paper appears technically sound, but there are few places that could be improved.

Line 122: it seems computationally expensive to calculate $\Sigma_{-k-k}^{-1}$, the inverse of a $(K-1)\times (K-1)$ matrix, for k=1,...,K, in each iteration, especially if $K$ is large. It is also not clear whether there are numerical stability issues to invert a large covariance matrix. More details could be provided.

Line 165: a conjugate normal inverse Wishart prior is applied. Is it possible to impose low rank assumption on the covariance matrix $\Sigma$ or sparse constraints on the inverse covariance matrix $\Sigma^{-1}$?

The authors argue that using the truncation of an infinite sum of gamma random variables to sample the PG random variable is not efficient and potentially inaccurate. However, Figure 1 (a) shows that the proposed approximate PG sampler is far from accurate. Moreover, although both the mean and variance of $z$, sampled in the way described in Line 206, match those of $z ~ PG(m,\rho)$, the sampled values could be negative, which should never be allowed.

According to Figure 1 (b) and Figure 6 (a), it seems that the accuracy of the PG sampler is neither critical for sampling $\eta_d^k$ from their correct posterior distributions nor important for obtaining low perplexity. I would suggest the authors to try the truncated version, which is very easy to implement and could be super fast if the truncation level is high, and compare it to the proposed approximate PG sampler. Note that a truncated PG sampler could also be compensated to match the mean of the exact PG sampler (Ref: the Appendix of M. Zhou, L. Li, D. Dunson and L. Carin. Lognormal and Gamma Mixed Negative Binomial Regression. ICML. 2012).

Originality and Significance:

Considering that the logistic-normal topic model is not new, and the Polya-Gamma latent variable based data augmentation techniques have been extensively studied for multinomial-logistic regression in Polson, Scott and Windle, 2013, the novelty of the paper appears to be a little bit thin.

The paper would be of particular interest to people who want to summarize the correlation between topics with a covariance matrix inferred from the data. But it is important to keep in mind that even if the most basic LDA is applied, which does not impose any correlations between topics in the prior, one may still discover the correlations between topics based on the posterior usages of topics between documents. Based on the experimental results, I am not convinced that why the logistical-normal topic model should be applied if one is interested to find correlation structures between topics, since simple analysis of the posterior topic proportion vectors of any topic modeling algorithms would be able to reveal which topics tend to appear together.
Summary: The paper applies the recently proposed Polya-Gamma data augmentation techniques to derive scalable inference for logistic-normal topic models. The novelty is a little bit thin but the empirical results are extensive.

Submitted by Assigned_Reviewer_7

The authors propose a method for scalably sampling from a logistic normal topic model. The trick they use introduces an auxiliary variable leveraging the Polya-Gamma distribution. They provide a natural parallel implementation.

Overall I found the paper easy to read despite not being very familiar with the Polya-Gamma distribution. And the paper does address the question of how to sample from these types of models whose use is perhaps limited by the complexity of sampling. However, I had questions around related work I was hoping the authors would have addressed.

The authors mention a few other works[7,8,13], and make an argument that these techniques "are not easy to sample from". It would be beneficial to have a clearer argument for why the authors' proposal is easier. Beyond a qualitative argument it would also be good to have a quantitative evaluation in Section 5 to see that the authors' approach is superior in some way.

In the experimental section the authors do compare to Y!LDA; it seems as though the implementation of Y!LDA is considerably faster (by up to an order of magnitude) with an indiscernable amount of improvement in perplexity. Does this call into question the utility of the approach?

Also I'd be curious to know a few more details about the parallel implementation. The authors intimate that mapreduce was used but a standard package such as Hadoop would incur substantial inter-iteration overhead (especially compared against the Y!LDA implementation).

nit: line 212: Appendix -> the appendix.
nit: Perhaps include a metric like cohesion since perplexity has been shown to not be representative of topic model fit.
Summary: The authors provide a novel approach to sampling from the CTM. The paper would be stronger if it were backed by more experiments against competing techniques.
Author Feedback

Author rebuttal: We thank the reviewers for acknowledging our contributions and providing valuable comments. We'll improve the paper and add more algorithm details. Below, we address the detailed comments.

To R4: robustness on other corpus sizes & numbers of topics:
We have similar observations; E.g., using various NIW priors (a ranges from 10 to 100,000) leads to similar performance on NYTimes (D=285K) when K=500. The number of sub-iterations (S) depends on K and average document length. Generally, more topics or longer documents require larger S; but it is fine to set S=1 or 2 for most experiments (D can be 6M; K can be 1000). We'll add more results.

To R6:
Q1: matrix inversion & numerical stability:
We're sorry for the misleading. In fact, we compute the conditional Gaussian using the precision matrix as in Eq.(2.96)(2.97) in (Bishop et al., Pattern Recognition and Machine Learning, 2006). Thus, only vector-vector product is involved in Eq.(2). Empirically, ~20% time is spent on sampling \eta.

For stability, we can effectively control the condition number of E_{NIW}[Sigma] by using appropriate priors. Let Q'= W'-W in line 173. Then Q' is PSD, and we have Q'=U^T \Lambda U, where U^T U = I and \Lambda=diag(\lambda). We used symmetric priors W=\kappa I=U^T \kappa I U in experiments. So, we have W'=U^T(\Lambda+\kappa I)U and E_{NIW}[\Sigma] = U^T (\Lambda+\kappa I)/(\kappa+D-K-1) U. By increasing \kappa, the condition number of E_{NIW}[\Sigma] (i.e., sqrt{(\lambda_{max}+\kappa)/(\lambda_{min}+\kappa)}) gets larger. (Note: \lambda_i >= 0 for PSD Q'). As in lines 256-258, we set \kappa=0.01D, which effectively avoided ill-conditioned matrices.

Q2: other priors:
It is an interesting suggestion. But an extensive discussion is beyond our scope since our main focus is on the scalable algorithm.

Q3: negative Polya-Gamma (PG) samples:
We never have negative samples since f(z) > sqrt{n/m} z > 0 if m < n, by substituting the expectations and variance ratio. (We noted a typo in line 209: E[y] should be divided by \rho).

Q4: Truncated PG sampler:
We'll add it (slower than our sampler to get comparable performance in various trials).

Q5: correlation structure by LDA & significance:
Though subsequent tools can calculate the correlation structure in LDA, it doesn't provide a unified solution. In contrast, CTM provides an elegant model that allows topics to interact closely; and this elegance also makes CTM suitable as building blocks for other sophisticated models (e.g., dynamic topic models [3]) (See Lines 33-46). Thus our scalable algorithm to CTMs would benefit other related models. The significance is agreed by R4.

To R7:
Q1: PG sampler & quantitative comparison:
Thanks for the suggestions. We'll add more explanations. Basically, as in Sec 3.1&3.3, sampling Z and (\mu, \Sigma) can be easily done. Thus, only sampling \eta (a Bayesian logistic regression (BayesLR) problem as in Sec 3.2) will distinguish different algorithms for CTM. Hence, we can indirectly compare the performance of various algorithms for CTM by comparing their samplers for BayesLR. [12] shows that the PG sampler outperforms several other methods in terms of effective sample size (ESS) per second, including the data augmentation (DA) algorithms in [OD04;FF10;GP12] and several Metropolis algorithms. [FF10] also claimed better performance than several other algorithms in ESS, including [7]. Given these results, we concluded that the PG sampler is the best algorithm to solve BayesLR, and thus integrated it for our scalable CTM algorithm.

We didn't include quantitative comparison to other DA algorithms for CTM because: 1) extensive comparison has been done for BayesLR as stated above; 2) no DA algorithms for CTM were publicly available; and 3) no parallel CTM algorithms were reported/evaluated. In fact, we contacted the authors but failed to get their implementation of the single-machine algorithm [8]. We plan to reproduce it and carry out direct comparison.

[OD04] S. O'Brien & D. Dunson. Bayesian multivariate logistic regression. Biometrics, 60:739-746, 2004.
[FF10] S. Fruhwirth-Schnatter & R. Fruhwirth. Data augmentation and MCMC for binary and multinomial logit models. In Statistical Modeling and Regression Structures, pp:111-132, 2010.
[GP12] R. Gramacy & N. Polson. Simulation-based regularized logistic regression. Bayesian Analysis, 7(3):567-90, 2012.

Q2: Comparison with Y!LDA:
As agreed by R4, the speed difference is brought by model difference: CTM extracts both topics and their correlation structure, whereas LDA only addresses the topic extraction task. Moreover, even having comparable perplexity as LDA, CTM is an important & challenging model, for which no scalable algorithms have been reported or carefully evaluated. Our work potentially makes CTM (and models built on it) applicable to large-scale applications.

Q3: Details about parallel implementation (MapReduce):
We're sorry for the confusion of MapReduce and will add more explanations. Basically, MapReduce doesn't refer to Hadoop/Google-MapReduce packages; instead, we mean the divide-conquer-reduce parallelization concept. In our case, it is implemented by IntelMPI with fully in-memory computation.

The parallelization strategy of CTM is twofold: 1) sample topic assignments and other local variables using the state-of-the-art asynchronous framework of Y!LDA; 2) sample global variables (\mu, \Sigma) from NIW using the above MapReduce procedures. Empirically, the NIW sampler (including MapReduce communication & computation) costs ~10% of the time (e.g., <20 seconds per iteration when K=500 on the 6M Wiki data), which is acceptable. Our scalability experiment (Sec 5.4) also shows that the overhead doesn't hurt system performance much. But since the overhead could be higher for larger clusters, developing a sophisticated asynchronous NIW sampler composes our future work.